# Pulmonary Surfactant in Adult ARDS: Current Perspectives and Future Directions

**DOI:** 10.3390/diagnostics13182964

**Published:** 2023-09-15

**Authors:** Ahilanandan Dushianthan, Michael P. W. Grocott, Ganapathy Senthil Murugan, Tom M. A. Wilkinson, Anthony D. Postle

**Affiliations:** 1National Institute for Health Research (NIHR) Southampton Biomedical Research Centre, University Hospital Southampton National Health System Foundation Trust, Southampton SO16 6YD, UK; mike.grocott@soton.ac.uk (M.P.W.G.); t.wilkinson@soton.ac.uk (T.M.A.W.); a.d.postle@soton.ac.uk (A.D.P.); 2Integrative Physiology and Critical Illness Group, Clinical and Experimental Sciences, Faculty of Medicine, University of Southampton, Southampton SO16 6YD, UK; 3Optoelectronics Research Centre, University of Southampton, Southampton SO17 1BJ, UK; smg@orc.soton.ac.uk

**Keywords:** ARDS, surfactant, phospholipids, DPPC, COVID-19, aerosolized

## Abstract

Acute respiratory distress syndrome (ARDS) is a major cause of hypoxemic respiratory failure in adults, leading to the requirement for mechanical ventilation and poorer outcomes. Dysregulated surfactant metabolism and function are characteristic of ARDS. A combination of alveolar epithelial damage leading to altered surfactant synthesis, secretion, and breakdown with increased functional inhibition from overt alveolar inflammation contributes to the clinical features of poor alveolar compliance and alveolar collapse. Quantitative and qualitative alterations in the bronchoalveolar lavage and tracheal aspirate surfactant composition contribute to ARDS pathogenesis. Compared to neonatal respiratory distress syndrome (nRDS), replacement studies of exogenous surfactants in adult ARDS suggest no survival benefit. However, these studies are limited by disease heterogeneity, variations in surfactant preparations, doses, and delivery methods. More importantly, the lack of mechanistic understanding of the exact reasons for dysregulated surfactant remains a significant issue. Moreover, studies suggest an extremely short half-life of replaced surfactant, implying increased catabolism. Refining surfactant preparations and delivery methods with additional co-interventions to counteract surfactant inhibition and degradation has the potential to enhance the biophysical characteristics of surfactant in vivo.

## 1. Introduction

### 1.1. Pulmonary Surfactant

Pulmonary surfactant is a complex mixture of phospholipids (80%), proteins (10%) and neutral lipids (10%). It is synthesized, secreted, and recycled by alveolar type-II epithelial cells (AT-II) with a primary function to reduce alveolar surface tension at the air–liquid interface, providing mechanical stability for gas exchange [1]. Surfactant components are also involved in innate immunity and are essential for the host’s defense mechanisms against infections [2,3]. The phospholipids account for most of surfactant composition with phosphatidylcholine (PC) and phosphatidylglycerol (PG) being the most abundant phospholipids. Minor phospholipids, including phosphatidylinositol (PI), phosphatidylethanolamine (PE), phosphatidylserine (PS), sphingomyelin (SM) and lysophosphatidylcholine (LPC) make up the rest of the phospholipid distribution [4,5]. The principal surface-active molecule is disaturated dipalmitoyl-PC or DPPC (PC16:0/16:0 or PC 32:0), which accounts for approximately 50% of PC [6]. Low surface tension is essential at the alveolar surface to minimize pressure gradients across the alveolar lining preventing premature airway collapse. Under dynamic compression, DPPC can reduce surface tension to near zero values in vitro [7].

There are four surfactant-based proteins, SP-A, SP-B, SP-C and SP-D. SP-B and SP-C are hydrophobic proteins involved in the adsorption of surfactant film, whereas SP-A and SP-D are hydrophilic and participate in innate immunity. Hereditary SP-B deficiency leads to lethal respiratory failure, whereas hereditary SP-C deficiency leads to acute and chronic lung diseases [8,9]. Animal models of SP-A deficiency can lead to increased susceptibility to respiratory tract infections, whereas SP-D knockout mice models demonstrate increased alveolar infiltration of macrophages, AT-II cell hyperplasia and excess phospholipid production leading to the development of emphysema [10]. Primary surfactant deficiency due to lung immaturity is the characteristic feature of neonatal respiratory distress syndrome (nRDS), where exogenous surfactant replacement is associated with improved clinical outcomes [11]. Detailed surfactant composition, metabolism and function are evaluated by excellent in-depth reviews [1,2,7,8,9,10,12,13].

### 1.2. Acute Respiratory Distress Syndrome (ARDS)

ARDS is a heterogenous disease process characterized by pathological changes of diffuse alveolar damage with alveolar epithelial and endothelial injury, leading to alveolar capillary leak and pulmonary oedema [14]. Clinically, patients present with poor alveolar compliance, non-hydrostatic pulmonary oedema, and hypoxemic respiratory failure [15]. It is an under-recognized syndrome even in an intensive care unit setting due to the complexity around the multi-component nature of its diagnostic definition, which has been evolving over the past 50 years [16]. According to the current Berlin definition, ARDS is diagnosed when there is an acute onset (<7 days) of symptoms, the presence of bilateral radiological opacities, varying severity of arterial hypoxemia (PaO_2_/FiO_2_ ratio < 100 mmHg: Severe, PaO_2_/FiO_2_ ratio: 100–200 mmHg as moderate, PaO_2_/FiO_2_: 200–300 mmHg as mild) and the absence of cardiogenic cause for pulmonary oedema [16].

The severity of the hypoxemia correlates with adverse outcome, where severe ARDS is associated with 40–50% mortality and the milder version with <30% mortality [16]. Sepsis from both direct and indirect lung injury is the primary risk factor for development of ARDS [17]. The mortality outcome is also variable between patients depending on the cause of ARDS. For instance, direct ARDS from pulmonary etiology associated with sepsis has a much higher mortality than-non-pulmonary ARDS without sepsis [18]. Moreover, patients with trauma related ARDS have a better prognosis than ARDS patients associated with cirrhosis or liver failure [19,20,21]. Treatment response is also vastly different between patients depending on their specific risk factors. Despite attempts to harmonize the ARDS diagnostic criteria to facilitate clinical trials, clinical heterogeneity remains a major issue [22]. Even in ARDS related to pulmonary etiology, there are variations in clinical outcomes and response from different insults such as viral infections, bacterial infections and chemical pneumonitis following aspiration of gastric contents. Recently, post hoc analyses of published ARDS randomized controlled trials suggest the existence of phenotypes according to the degree of lung and systemic inflammation as hyper-inflammatory and hypo-inflammatory with variations in responses to treatment [23].

## 2. Surfactant Abnormalities in ARDS

Surfactant compositional and functional abnormalities in ARDS are likely due to several reasons. The pathophysiological processes leading up to ARDS are complex and involve alveolar epithelial cellular apoptosis with significant neutrophil-mediated inflammatory infiltration, pulmonary oedema, and invasion of alveolar space by plasma and inflammatory constituents. Earlier studies on ARDS patients identified impaired surface film compressibility from lavage fluid from ARDS patients [24]. Although surfactant abnormalities are consistently seen in ARDS patients, whether they are the primary cause of lung injury, or a consequence of the initial insult is not fully understood. Several studies have characterized surfactant molecular composition and alterations in ARDS patients, which are detailed below. A summary of surfactant alterations and pathophysiological consequences in ARDS are schematically presented in Figure 1.

### 2.1. Lung Fluid Phospholipid Alterations in ARDS

ARDS patients exhibit qualitative and quantitative changes in the phospholipid composition of lung fluid recovered by bronchoalveolar lavage or tracheal aspiration. Although the total phospholipid content is variable among studies, the measurements of absolute phospholipid concentrations are limited by variability in sample recovery and analytical methods. The most common finding of surfactant alteration is related to the molecular compositional variations in phospholipid distribution. The first comprehensive surfactant phospholipid analysis in human ARDS, Hallman et al. in 1982, demonstrated low levels of lecithin (phosphatidylcholine), particularly the disaturated lecithin (disaturated PC, predominantly DPPC) and PG in lavage fluid from ARDS patients. In comparison, the relative concentrations of SM, PS, and PI fractions were much higher. Moreover, low levels of lecithin/sphingomyelin ratio (<2) and PG (<1% of total phospholipids) were consistently associated with respiratory failure [25].

Following these findings, a study evaluated lavage fluid phospholipid composition in trauma-related ARDS and classified patients according to the severity of respiratory failure [26]. This study demonstrated a correlation progressive decrease in PC composition and severity of respiratory failure, suggesting that the lower fractional PC composition is related to the severity of ARDS [26]. When trauma patients with respiratory failure developed sepsis, there were significant perturbations in phospholipid distribution in the alveolar fluid, with a substantial increase in PE combined with lower levels of lavage PC [27]. The surfactant biophysical perturbations of altered surface activity, low PC and PG with increased PI, SM, PS and PE are also seen in patients at-risk of developing ARDS, suggesting that early surfactant supplementation may mitigate ARDS progression [28].

The total lavage phospholipid (PL) content is variable between studies. There are many reasons for this variability, including various degrees of inflammatory cell membrane infiltration, and variations in the sample recovery and analytical methods used [25,28,29]. The lavage PL content may also depend on when the phospholipid analysis was performed during the various stages of ARDS. When ARDS is classified as early (<36 h after clinical features and diagnosis of ARDS), intermediate (>36 h, <6 days), or late (>6 days), the total PL content is increased in the early stages, but with a marked reduction in concentrations at the later stages of ARDS [30]. It is important to recognize that recovered lung fluid from ARDS patients contains membrane material other than lung surfactant, predominately extracellular vesicles from the increased airway neutrophil concentration. Consequently, the increased fractional concentrations of SM and PS, characteristic of cell membranes, are most likely derived from extracellular vesicles rather than from altered surfactant composition.

A more detailed analysis of the fatty acid profile of lavage fluid showed a marked reduction in palmitic acid (16:0) and saturated fatty acids in patients with ARDS [31]. Consistent with this finding, molecular species analysis of lavage phospholipid from ARDS patients demonstrated a significant reduction in dipalmitoyl PC and increased fractional concentrations of unsaturated and polyunsaturated PC species such as PC16:0/18:2, PC16:0/18:1 and PC16:0/20:4, characteristic of cell membrane material [32,33]. The degree of oxygenation impairment correlated with DPPC levels in large aggregate fractions isolated by high-speed centrifugation and more importantly, continued phospholipid alterations were associated with adverse outcomes [32]. Consistent with this finding Nakos et al. demonstrated that lack of recovery of lavaged PC during the disease course of ARDS is associated with increased mortality [30]. This suggests that during surfactant replacement a longer duration of therapy may be required in some patients with continued surfactant deficiency.

More recently, similar to the findings of the ARDS population, significant alterations in surfactant phospholipid molecular composition with reduced PC, DPPC and PG levels with reciprocal increments in other phospholipids, were seen in COVID-19 patients with severe pneumonia and ARDS [34,35]. In summary, alveolar lavage fluid from ARDS patients demonstrates significant alterations in the functional ability to maintain surface tension, phospholipid content, distribution of phospholipid categories, and PC molecular distribution, and continued surfactant alterations are associated with adverse clinical outcomes.

### 2.2. Surfactant Protein Alterations in ARDS

Alterations in surfactant composition are not limited to the phospholipid fraction. Although the findings are variable between studies, in general, there are reductions in concentrations of lavage surfactant proteins SP-A, SP-B and SP-C with reciprocal increases in plasma SP-A and SP-D levels [28,32,36]. While serum SP-A and SP-B levels may predict the development of ARDS [37,38], higher plasma SP-D levels correlate with disease severity and poor outcomes [39]. These high concentrations in plasma reflect the increased alveolar permeability and the consequent leakage of SP-D into the systemic circulation.

### 2.3. Surfactant Extraction and Analytical Methods

Alveolar surfactant isolation, purification and quantification require invasive procedures such as bronchoalveolar lavage (BAL), which, although a safe procedure, requires medical personnel to perform the procedure. Patients often require additional sedation, and in mechanically ventilated patients with hypoxemic respiratory failure desaturations and change in respiratory mechanics during and after the procedure are common [40,41]. Moreover, there are significant variations in quantitative measurements due to variations in sample recovery and will depend on the total segments lavaged. A theoretical risk of further surfactant depletion and atelectasis following a large volume lavage also exists. Nevertheless, BAL has been extensively used in ARDS patients to access surfactant material. Small volume BAL is an alternative to characterize surfactant composition without further compromising the patient’s clinical condition, but limits the ability to perform quantitative measurements [25,33]. Similarly, tracheal aspirates can also minimize procedure-related complications. Studies of healthy humans suggest comparable phospholipid composition from tracheal aspirates and can be used for surfactant molecular analysis [5]. However, the limitations include variability in recovery and inability to provide quantitative measures and, in ARDS patients, the phospholipid composition may be contaminated by inflammatory cell membrane phospholipid constituents [33]. Recent advances in microparticle extraction from lungs utilizing Particles of Exhaled Air (PExA) is an alternative and attractive way to extract alveolar surfactant material non-invasively [42].

Surfactant analysis requires centrifugation to extract the surfactant pellet, followed by lipid extraction and analysis by various gas–liquid chromatography, high performance liquid chromatography (HPLC) and mass spectrometry (ESI-MS) techniques to quantify surfactant phospholipids. De novo surfactant synthesis and metabolism in humans can be characterized by isotope labelling of surfactant phospholipid components. A combination of isotope labelling with tracer kinetics modelling and mass spectrometry analytical methods is used to measure surfactant synthesis and metabolism in ARDS patients in vivo [33,43]. The tracer substances vary between studies, but essentially include deuterated choline, deuterated water, 13C-glucose, 13C-palmitate, and 13C-acetate, which all incorporate into surfactant phospholipids enabling assessment of synthesis and metabolism of endogenous surfactant de novo [44,45]. Recent advances in spectroscopic techniques can minimize the analytical time required to measure specific surfactant phospholipid components such as DPPC, bypassing the need for detailed mass spectrometry analytical steps [35,46].

## 3. Molecular Mechanisms of Surfactant Alterations in ARDS

The molecular mechanisms of surfactant alterations in ARDS are complex. ARDS is characterized by significant inflammatory cell infiltration and alveolar epithelial and endothelial injury. Lung infection and aspiration of gastric contents can directly damage AT-II cells and impair surfactant synthesis, secretion, and recycling. Studies of various animal models of lung injury and AT-II cells suggest variations in surfactant synthesis [47,48,49,50,51]. The conflicting results are due to the variability in the lung injury models, the dose and duration of the insult exposure, and the timing of surfactant measurements taken. Human adult studies of isotope labelling of surfactant precursors suggest that despite very low surfactant PC, DPPC, or SatPC pool sizes, there may be increased synthesis and secretion by existing AT-II cells [33,43]. This implies that surfactant synthesis may be preserved, or even increased in functional AT-II cells and other factors may contribute to the surfactant alterations seen in ARDS. Both direct and indirect injuries can increase alveolar and systemic inflammatory response leading to cellular damage. Alveolar endothelial and epithelial injury can cause an influx of protein-rich pulmonary oedema, containing inflammatory exudate, cellular debris, and plasma proteins, which can destabilize surfactant film and directly impair surfactant activity [52,53,54,55]. Increased oxidative stress from overt inflammation and alveolar hyperoxia from oxygen therapy can result in the oxidation of surfactant phospholipids and proteins [56,57]. Moreover, activation of sPLA2-mediated hydrolysis leads to surfactant phospholipid catabolism and generation of lysophosphatidylcholines, compromising the surfactant function even further [58,59,60]. While all these mechanisms can lead to alterations in surfactant composition and function, assessing relative contribution is far more complex, particularly in human in vivo clinical settings.

## 4. Surfactant Replacement in ARDS

Surfactant replacement has been evaluated as a potential therapeutic target for ARDS for the past few decades. Even though the exact cause of surfactant abnormalities in ARDS is not known, nor how dysregulated surfactant directly leads to adverse clinical outcomes, both primary surfactant deficiency from direct alveolar epithelial injury and secondary abnormalities of surfactant functional inhibition from endothelial leakage are likely to contribute to the ARDS pathogenesis. Consequently, replacement with an exogenous surfactant may restore surfactant homeostasis and alveolar epithelial lining integrity. While surfactant replacement is the standard of care to improve outcomes in premature neonates with primary surfactant deficiency, studies of adult patients with ARDS so far have demonstrated no survival benefits. Currently, randomized controlled trials do not support the routine use of surfactant replacement and the sound scientific rationale that originated from preclinical studies of animal models of surfactant depletion and deficiency requires further refinement.

### Surfactant Replacement Clinical Trials

Following several observational animal and phase 1 human studies proposing beneficial physiological effects of exogenous surfactant in ARDS, a large randomized controlled trial of synthetic aerosolized surfactant (Exosurf) on sepsis induced moderate ARDS patients demonstrated no survival benefits [61]. In this study of 725 ARDS patients, the surfactant was aerosolized continuously for 5 days. Although the dose of surfactant delivered was thought to be high enough to counteract the effect of surfactant inhibition from sepsis related lung injury, it was estimated that only around 5% of surfactant was delivered effectively into the lungs. Moreover, the absence of surfactant proteins in the surfactant preparation may have contributed to the lack of clinical efficacy [61].

Gregory et al. performed a small (*n* = 59) randomized controlled trial investigating the effect natural bovine lung surfactant delivered via an endotracheal catheter in patients with ARDS. The surfactant dose was either 50 or 100 mg/kg instilled in large volumes (4 mL/kg) four or eight times a day [62]. This proof of principle and feasibility study established the mode of surfactant delivery beyond aerosolization. Subsequent several small non-controlled studies of bronchoscopy administration of 50–500 mg/kg surfactant dose of different types, both natural and synthetic surfactants have shown beneficial effects in improving surfactant function, respiratory gas exchange and hemodynamics [63,64,65,66,67,68] (Table 1).

Following successful animal studies [69,70,71,72], several randomized controlled trials investigated the effect of a recombinant surfactant protein-C-based surfactant (rSP-C, Venticute) in ARDS. The first phase I/II trial gave either 1 mL/kg of rSP-C or 0.5 mL/kg of rSP-C surfactant four times in the first 24 h via an endotracheal catheter. This small study of 40 patients demonstrated the safety and feasibility of rSP-C delivery in the ARDS population [73]. However, surfactant biophysical function was not preserved beyond 48 h after the last dose, and no exogenous surfactant was detected beyond 120 h, highlighting the need for prolonged replacement [73]. Replicating this methodology with 1 mL/kg of rSP-C, a subsequent worldwide multi-center randomized controlled trial of 448 moderate ARDS patients showed improvements in oxygenation without any significant mortality benefits. In this study, sepsis was the most predisposing event (50%), followed by pneumonia (30%) and patients with direct lung injury compared to indirect lung injury had better survival outcomes following exogenous surfactant replacement [74]. Subsequent post hoc, subgroup pooled analysis of rSP-C trials showed improved oxygenation and reduced mortality in those with direct lung injury with either pneumonia or aspiration compared to those with other causes of ARDS [75].

The positive findings from the post hoc analysis led to the design of the largest multi-center rSP-C surfactant trial to date [76]. This trial of 843 direct ARDS patients was stopped early due to futility and failed to show any improvement in oxygenation or survival outcomes. The lack of gas exchange improvement contradicted the results from previously published rSP-C trials. The trial design was different from the previous trials in several ways. (1) The study only included patients with direct lung injury. (2) Patients were treated with 1 mL/kg LBW rSP-C up to eight doses for four days or whilst they remained intubated. (3) A shearing step was introduced during surfactant preparation to improve dispersion. Despite including a more homogenous patient group and prolonged surfactant replacement, the introduction of the shearing step may have contributed to the lack of clinical efficacy. The authors concluded that the shearing step introduced to improve dispersion, combined with the introduction of air, led to demulsified surfactant that lacked surface-lowering activity and increased susceptibility to surfactant inhibition from plasma constituents such as fibrinogen [76].

A study of a large bolus of natural freeze-dried porcine surfactant (HL 10) cumulative dose of 600 mg/kg delivered as two 300 mL syringes of three doses administered endotracheally in ARDS patients was also stopped early due to futility [77]. The common etiology of ARDS was sepsis, followed by pneumonia. There was an increased transient hypoxemia (oxygen saturation of <88%) and hypotension associated with the delivery of a large bolus of surfactant. The intervention group had a trend toward increased mortality. However, the separation in mortality was only seen around 21 days after surfactant administration, suggesting the outcomes may not be directly related to the surfactant intervention [77]. The lack of efficacy may be related to the surfactant preparation. The absence of published preclinical animal or phase I/II human data on the effect of HL-10 on injured lung models preceding this trial makes it difficult to translate the biophysical properties of the surfactant preparation in vivo [78].

The effect of Calfactant, a natural surfactant extracted from calf lung wash was assessed in ARDS patients due to direct lung injury. Up to three doses of 30 mg of Calfactant per centimeter height did not demonstrate any clinical benefits, including oxygenation variables [79]. However, the shorter duration of study intervention may be a limitation of the study design. A meta-analysis combining all these trials confirmed the findings of no survival benefits and lack of oxygenation improvement [80]. The surfactant ARDS trials are listed in Table 2.

## 5. Surfactant Replacement-Unanswered Research Questions

Currently, there is no evidence for the routine use of surfactant in ARDS. While it is possible to claim that exogenous surfactant is an ineffective treatment strategy for ARDS patients, several questions remain unanswered. (1) Which patients require surfactant therapy and what type of surfactant is indicated? (2) What is the ideal dose? (3) What is the ideal delivery mode? (4) What is the fate of the delivered surfactant? (5) What proportion of the supplemented surfactant is surface-active in vivo? (6) How to optimize surfactant inhibition and breakdown? Future trials should aim to stratify patients according to surfactant phenotypes (e.g., reduced synthesis, increased breakdown, etc.). Moreover, further exploration of detailed mechanisms of surfactant metabolism to identify co-supplementation of potential therapies to minimize surfactant breakdown by hydrolysis, oxidation and inhibition may help refine this intervention further.

### 5.1. Surfactant Preparation: Synthetic vs. Natural Surfactant Replacement

The optimal surfactant replacement strategy in ARDS is not fully defined. Nevertheless, it remains unclear if modifications in surfactant preparations or dose and duration of therapy may influence clinical outcomes. Natural surfactants both derived from animal lungs (porcine and bovine) and synthetic preparations are available in the market for the use in nRDS. The natural surfactants appear to be superior to the protein free synthetic surfactants [82]. The presence of surfactant proteins will help to facilitate spreading and stabilization of surfactant film at the air–liquid interface. However, natural surfactant is very expensive and large quantities may be required to counteract the effect of surfactant inhibition in adults. Moreover, there are compositional variations between animal natural and synthetic surfactants [83]. A meta-analysis of neonatal trials comparing natural versus synthetics surfactant preparations concluded that natural surfactant is superior in comparison to its synthetic counterparts [82]. However, earlier synthetic preparations lacked surfactant proteins and recent advances in synthetic surfactant preparations containing surfactant protein analogs suggest comparable efficacy [84,85].

### 5.2. The Mode of Surfactant Delivery

An effective airway surfactant delivery is essential for homogenous distribution of optimal surfactant dose to the alveolus. So far, all adult surfactant replacement studies have been conducted on mechanically ventilated ARDS patients and the surfactant was delivered through three distinct modes: aerosolization, intratracheal delivery, and bronchoscopic delivery. Although earlier studies used the aerosolization technique, they were limited by poor alveolar deposition [61,81]. Few non-controlled studies have employed bronchoscopic surfactant delivery and documented favorable outcomes (Table 1). However, bronchoscopic delivery can be labor exhaustive and require specialist experts to perform the procedure. Consequently, most of the studies which used bronchoscopic delivery were uncontrolled, non-randomized clinical studies, and large clinical trials are lacking. The standard method of surfactant administration is intratracheal instillation, which has been employed by both adult and neonatal populations without any significant adverse events. Nevertheless, a study of large volume intratracheal surfactant administration was associated with transient hypoxemia and hypotension [77]. Moreover, intratracheal administration requires patients to be endotracheally intubated and is not applicable for surfactant delivery via non-invasive ventilation (NIV) or high flow nasal oxygen (HFNO).

Recent advances in aerosolization technology with photo-defined aperture plate (PDAP) in combination with breath-synchronized nebulization only delivering during the first 80% of the inspiratory phase enables more efficient surfactant delivery [86]. This may also have the advantage of delivering therapies early and can be instituted prior to the development of ARDS. Moreover, this mode of delivery can be used in spontaneously ventilating patients, during non-invasive therapies such as NIV/CPAP or HFNO as a prophylaxis and potential to prevent disease progression, which has not been explored in adult ARDS before. Scintigraphy studies have shown that the vibrating mesh technology is much more effective in distal lung deposition than traditional jet nebulization [87]. The feasibility of these techniques has been evaluated in a study of COVID-19 adult patients requiring non-invasive ventilation and in premature neonates with 26–30 weeks of gestation [88,89].

Several factors influence intrapulmonary surfactant distribution. Beyond the surfactant preparation characteristics and the surfactant volume and viscosity, the degree of non-homogenous aeration of the lungs may alter surfactant delivery. Human studies exploring surfactant distribution during surfactant replacement are lacking. However, predefining the degree of atelectasis by the utility of an ultrasound (USS) or CT scan may help target regional surfactant therapy where it is most needed. This may also prevent surfactant delivery to relatively normal lung areas and minimize overdistention. Although lung USS and the lung ultrasound score (LUS score) has been evaluated as a tool to identify neonates requiring surfactant treatment, so far, adult studies have not explored this proof-of-concept [90].

### 5.3. Surfactant Delivery via High Flow Nasal Oxygen and Non-Invasive Ventialtion

Administration of exogenous surfactant early in the disease process may prevent continued deterioration and requirement for mechanical ventilation, minimizing secondary lung injury from hyperoxia and mechanical ventilation may avoid ARDS progression. The first such report was from premature nRDS patients who required continuous positive airway pressure (CPAP). The surfactant was aerosolized during spontaneous respiration, and it was feasible to deliver surfactant via non-invasive ventilation devices with improvements in oxygenation [91]. Since then, multiple clinical trials of premature neonates have been published in this area. Recent studies of aerosolized surfactant in preterm infants during CPAP suggest it is feasible to deliver surfactant while on CPAP without any safety concerns and may reduce the need for intubation and mechanical ventilation [89,92]. While early surfactant therapy in combination with CPAP appears to be feasible in neonates, adult studies are needed to assess the efficacy of exogenous surfactant as a prophylactic measure to prevent development of ARDS in mitigating the need for subsequent mechanical ventilation, which is associated with significant morbidity and mortality. The advances in aerosolized non-invasive surfactant delivery in pre-term infants may help to evaluate surfactant replacement as a preventative strategy to minimize the progression and burden of ARDS, which requires further exploration in the adult population.

### 5.4. The Dose of Surfactant

There is significant variation in the dose of surfactant delivered in adult ARDS clinical trials. The generally accepted practice is to give large quantities of surfactant to prevent surfactant inhibition from pulmonary oedema and plasma constituents. The typical dose given for nRDS is 100–200 mg/kg and studies vary depending on surfactant preparation [93]. However, translating these doses for a typical adult of 70 kg suggest larger doses (3.5–7 g) are required, which can be expensive. Nevertheless, a range of doses from 25–300 mg/kg have been used in adult studies to counteract surfactant inhibition (Table 2).

Instilled surfactant dose volume is a critical factor for success. The adult lung has significantly more conducting airways and conducting surface area for coating loss [94]. A higher volume or multiple doses may help the surfactant reach the distal alveolus better. However, while a larger volume of surfactant is better for surfactant distribution, it may also have detrimental effects during delivery, predisposing to hypoxemia and cardiovascular compromise [77].

In summary, while endotracheal instillation remains the primary delivery method in most clinical trials, further detailed mechanistic studies are needed to evaluate novel aerosolization techniques, particularly in non-ventilated spontaneously breathing patients to prevent disease progression.

### 5.5. Pharmacokinetics of Replaced Surfactant and the Duration of Surfactant Therapy

There are insufficient data on the pharmacokinetics of replaced surfactant, particularly in adults with ARDS. This is related to the inability to rapidly measure the fate of replaced natural surfactant compounds from various compartments. In nRDS, when to provide a repeat dose of surfactant is often clinically judged from either a lack of improvement or further deterioration in oxygenation and ventilation variables, which are often delayed signs in predicting the need for additional and ongoing replacement. Moreover, additional dose calculations are often estimated as the response is dependent on several factors.

In nRDS randomized controlled trials, multiple surfactant dosing is more effective than single dosing [95,96,97]. The dosing is essentially based on guidance provided by the manufacturers based on preclinical animal studies and not from in vivo data from real-world human injury models of ARDS. Repeat dosing is offered when there is lack of response or worsening of respiratory failure and is not an ideal method of assessing the suitability for a repeat dose. Isotope-labelled disaturated phosphatidylcholine (DSPC) has been used to measure pharmacokinetics in patients with nRDS. These tracer studies of nRDS have shown that there are significant variations in the surfactant DSPC half-life and the shorter half-life in some patients is thought to be due to additional lung injury and inflammation [98,99,100,101,102]. Consistent with these findings, previous ARDS studies have shown that the augmentations of surfactant phospholipids and the surface reducing physical properties of replaced surfactant in vivo is short-lived [63,73]. A recent study of COVID-19 patients also showed a very short half-life of supplemented aerosolized bovine surfactant, suggesting multiple and prolonged duration of therapy may be required for sustained clinical response [34]. However, these measurements require extensive laboratory processes including lipid extraction and mass spectrometry analytical methods and are not available as a point of care technique to influence rapid clinical decisions. Development of rapid analytical techniques may help to make decisions regarding initial dosage, the need for repeated doses and duration of therapy to improve clinical outcomes [46].

### 5.6. Heterogeneity of ARDS Patients and Treatment Effect

ARDS is a heterogenous disease process encompassing a multitude of etiological insults from various causes. Most surfactant replacement studies included a mixed cohort of ARDS patients, including sepsis, pneumonia, trauma, burns, pancreatitis, and aspiration (Table 2). While it is intuitive to replace surfactant in all groups of patients with ARDS, the pathogenesis is likely to be more complex and variable between different insults. Some patients may have surfactant deficiency due to reduced surfactant synthesis and metabolism from alveolar epithelial cell death [103]. In contrast, others may have primary surfactant inhibition due to increased endothelial permeability from systemic causes or increased surfactant breakdown from overt inflammation leading to activation of hydrolytic and oxidative pathways [56,104]. However, there is likely a combination of all these factors perpetuating surfactant deficiency and this clinical heterogeneity needs to be addressed by clinical trials prior to exogenous surfactant supplementation.

The etiology and severity of ARDS may influence the outcome after surfactant supplementation. Earlier studies conducted on septic ARDS patients had several issues, including poor alveolar deposition from the aerosolization of surfactant and lack of surfactant proteins, making it difficult to make any conclusions. So far, all ARDS clinical trials are conducted on patients with moderate ARDS. The post hoc analysis of rSP-C clinical trials suggests that severe direct lung injury from pneumonia or aspiration of gastric contents may be more responsive to surfactant replacement than indirect causes such as sepsis [75]. The most severe patients defined by the PaO_2_/FiO_2_ ratio of up to 100 mmHg had the most improvement in oxygenation from surfactant replacement [75]. However, despite this positive post hoc analysis of rSP-C studies showing improved outcomes in patients with direct lung injury, subsequent larger surfactant replacement studies of ARDS patients with direct lung injury failed to show any survival benefits. However, these studies were limited by their methodology. In one study, introducing the additional step to improve surfactant spreading may have compromised the surfactant integrity; in the other, the surfactant was delivered for a short duration [76,78]. Future studies should explore prolonged exogenous surfactant replacement focusing on severe direct lung injury traits.

In comparison to the premature neonates with nRDS, adult ARDS trials have failed to identify a specific target group who may benefit from exogenous surfactant replacement. Similarly, the effect of surfactant replacement in pediatric ARDS are limited [105,106]. This difference in the response to exogenous surfactant replacement between nRDS and other disease cohorts may be due to the complexity of pathological processes including the progressive development of immune system leading to inflammation mediated increased surfactant catabolism in adult and pediatric ARDS. Apart from giving larger doses, studies have failed to address the impact of surfactant inhibition and increased breakdown. High concentrations of inspired oxygen are often required to treat ARDS, directly exposing alveolar epithelial cells and surfactant material to alveolar hyperoxia. Alveolar hyperoxia can oxidase surfactant phospholipids, making them inactive [107,108]. Moreover, increased secretory phospholipase A_2_ (sPLA_2_) is a prominent feature of ARDS, and sPLA_2_-mediated surfactant hydrolysis may also increase surfactant phospholipid breakdown, compromising its surface-active ability [58,59,60]. Although large doses of exogenous surfactant are often given to mitigate surfactant inhibition by the alveolar inflammatory milieu, the supplemented surfactant is likely at risk of enduring the same fate as the endogenous surfactant. Although surfactant preparations with surfactant proteins are better at resisting surfactant inhibition, additional co-interventions may be required to minimize the supplemented surfactant’s inhibition, oxidation, and hydrolysis. Characterization of patients according to surfactant metabolism with consideration of optimizing surfactant preparations to resist surfactant inhibition and co-interventions to minimize rapid catabolism may enhance surfactant activity in vivo and enable individualized targeted surfactant therapy in the future [109,110,111].

## 6. Pulmonary Surfactant as An Antioxidant to Minimize Oxidative Damage

Surfactant has antioxidant and anti-inflammatory properties. Alveolar hyperoxia is inevitable in certain circumstances where there is significant shunt preventing adequate gas exchange presenting as severe hypoxemic respiratory failure and patients are treated with an increasing amount of inspired oxygen to mitigate systemic hypoxemia. The recent COVID-19 pneumonia-related ARDS is a classic example. While alveolar hyperoxia and oxidative lung injury can lead to both endogenous and exogenous surfactant dysfunction, an exogenous surfactant may act as an antioxidant to mitigate hyperoxia-induced lung injury [112,113,114]. Although the antioxidant capacity may vary between surfactant preparations, natural surfactants contain large amounts of superoxide dismutase (SOD) and catalase (CAT) and can mitigate oxidative lung injury [115]. Animal studies of natural surfactant replacement supplemented with SOD and CAT can moderate oxidative lung injury and improve gas exchange [116]. The addition of corticosteroids in the form of beclomethasone to natural surfactants also seems to reduce the lung oxidative stress as measured by total hydroperoxide and oxidation protein products [117].

## 7. Can Alveolar Biomarkers Help to Identify Surfactant Deficiency vs. Surfactant Inhibition and Requirement for Multiple Dosing during Replacement

In a study by Hallman et al., an L/S (lecithin to sphingomyelin) ratio of <2 and a PG < 1% of glycerophospholipids is associated with 100% acute respiratory failure [25]. In a cohort of polytrauma patients, the surfactant fractional phosphatidylcholine composition is inversely correlated with respiratory failure score (i.e., the higher the respiratory failure score, the lower the concentrations of alveolar PC), implying that lower levels of PC is associated with the severity of respiratory failure. Moreover, lower PC composition with a lack of recovery during the ARDS course is associated with increased mortality [30]. While these studies imply that lower PC and DPPC levels are associated with altered minimum surface-tension-reducing characteristics of surfactants, further larger studies are needed to evaluate the direct correlation of dysregulated surfactant molecular composition with diagnosis, disease progression and outcomes in ARDS. Refining surfactant extraction and analytical methods to quantify surfactant variables at the point of care may enable rapid assessment pre- and post-intervention to identify surfactant deficiency and inform ideal dosages and the frequency of treatment. While mass spectrometry is commonly employed for quantifying surfactant phospholipids, vibrational spectroscopy has been proposed as an alternative rapid tool to evaluate surfactant phospholipids, bypassing the need for laborious mass-spectrometry-based analytical methods [35,46].

## 8. Future Directions

Surfactant therapy in ARDS remains elusive, and despite positive results from pre-clinical studies, the positive outcomes were not replicated by large randomized controlled trials. However, studies suggest that there is an increased exogenous surfactant turnover, implying rapid catabolism of the replaced surfactant. Targeted measures are required to minimize the increased surfactant inhibition and breakdown. Alveolar hyperoxia causing oxidative catabolism of surfactant phospholipids and/or increased activation of hydrolytic pathways by secretory phospholipase A_2_-mediated mechanisms require further exploration. Moreover, modification of surfactant preparations to resist inhibition and breakdown with detailed assessment of surfactant integrity and spreading characteristics in human in vivo models may help design future clinical studies. Early treatment in spontaneously ventilating patients via high flow nasal oxygen or non-invasive ventilation may prevent ARDS progression, which has not been assessed before and requires further evaluation. Moreover, future clinical trials of exogenous surfactant replacement should include lung protective protocolized ventilation strategies to harmonies ventilation settings across centers to optimize delivery and minimize additional ventilation-induced lung injury.

## 9. Conclusions

ARDS is a significant clinical burden, characterized by alterations in alveolar surfactant composition with reduced levels of phospholipids, DPPC, PG surfactant proteins and compromised biophysical function. The detected BAL and tracheal aspirate changes in the surfactant composition are likely due to inflammatory cellular membranes and extracellular vesicles in combination with alterations in surfactant metabolism. Pulmonary surfactant metabolism is a complex, highly regulated mechanism that requires a finite balance of surfactant synthesis, secretion, and recycling to maintain alveolar requirements. Overt direct and indirect inflammation alters this balance leading to increased surface tension, alveolar collapse and poor lung compliance exacerbating the initial insult. Although there are several randomized controlled trials of exogenous surfactant replacement in ARDS, these are largely ineffective in improving clinical outcomes. This may be due to several reasons. The earlier aerosolization surfactant studies were limited by poor alveolar distribution. The post hoc results from rSP-C studies suggest that patients with severe direct lung injury may be more responsive to surfactant replacement. However, a subsequent large trial failed to show any mortality benefits, but was limited by its methodology. Disease heterogeneity and a lack of understanding of the exact reasons for dysregulated surfactant remain significant issues in designing clinical trials. The fate of replaced exogenous surfactant in vivo is unknown and needs closer scrutiny to minimize rapid turnover. Non-invasive and rapid bedside techniques may help refine surfactant analytical methods. Improving surfactant preparations, doses, and delivery methods with considerations of adding co-interventions to withstand inhibition and breakdown may enhance surfactant biophysical properties in vivo. Moreover, the recent advances in non-invasive aerosolization delivery methods suggest effective delivery and should be considered as a preventative tool to reduce ARDS progression in patients requiring high flow nasal oxygen and non-invasive ventilation.

## Figures and Tables

**Figure 1 diagnostics-13-02964-f001:**
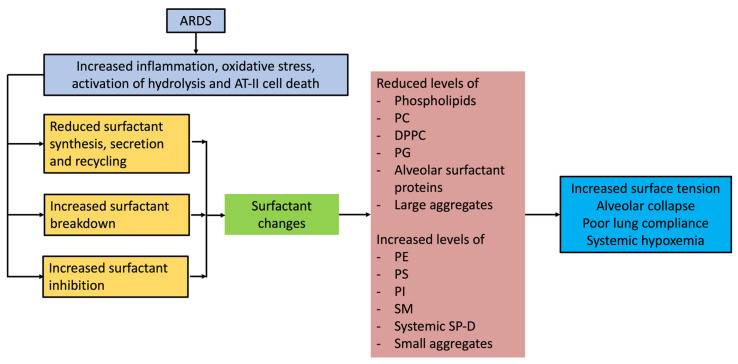
Surfactant alterations and the pathophysiological consequences in ARDS.

**Table 1 diagnostics-13-02964-t001:** The bronchoscopic surfactant delivery studies of ARDS.

Study	Number of Patients	Delivery Method	Dose of Therapy	Type of Surfactant	Outcomes
Spragg RC 1994 [63]	N = 6 Mix ARDS Cross over design	Bronchoscopy	Single dose 50–60 mg/kg	Natural porcine surfactant	Feasible to deliver No increased proinflammatory activity Increment in PL content after 3 h and fell by 24 h Variation in inhibition of surfactant function between patients
Walmrath D 1996 [64]	N = 12 Septic ARDS	Bronchoscopy	Single dose of 300 mg/kg Second dose of 200 mg/kg if improvement not sustained	Natural bovine surfactant (Alveofact)	Improvement in oxygenation
Wiswell TE 1999 [65]	N = 12 Mix ARDS	Bronchoscopy	3 groups 30 mL/segment (2.5 mg/mL) + 30 mL/segment (10 mg/mL) or 60 mL/segment of 2.5 mg/mL + 30 mL/segment (10 mg/mL) Or 60 mL/segment of 2.5 mg/mL + 30 mL/segment (10 mg/mL) + repeat dosing (6–24 h)	Synthetic KL4 peptide (DPPC + POPG + palmitic acid)	Safe and feasible to deliver Improvement in FiO_2_ and PEEP
Walmrath D 2002 [66] Gunther A 2002 [67]	N = 27 Septic ARDS	Bronchoscopy	Single dose of 300 mg/kg Second dose of 200 mg/kg if improvement not sustained	Natural bovine surfactant (Alveofact)	Improved oxygenation Improved biochemical and biophysical properties Increased BAL PL content Improved PC and PG
Tsangaris I 2007 [68]	N = 16 Trauma and severe refractory hypoxemia RCT	Bronchoscopy	Single dose of 200 mg/kg	Natural bovine surfactant (Alveofact)	Improved oxygenation Improved compliance

ARDS: acute respiratory distress syndrome; BAL: bronchoalveolar lavage; DPPC: dipalmitoyl phosphatidylcholine; PC: phosphatidylcholine; PEEP: positive end expiratory pressure; PG: phosphatidylglycerol; PL: phospholipid; POPG: palmitoyloleoyl phosphatidylglycerol; RCT: randomized controlled trial.

**Table 2 diagnostics-13-02964-t002:** The prominent randomized controlled trials of exogenous surfactant in ARDS.

Study	Cohort	Surfactant Preparation, Doses, and Delivery Methods	Outcome	Potential Issues
Weg 1994 [81]	N = 51 Sepsis induced ARDS	Synthetic surfactant (Exosurf) Aerosolized Dose: 21.9 mg or 43.5 mg DPPC/kg/day Continuous Duration: 120 h	Feasible and safe to deliver Trend towards lower mortality	Poor alveolar deposition Only included sepsis induced ARDS patients No surfactant proteins
Anzueto 1996 [61]	N = 725 Sepsis induced ARDS	Synthetic surfactant (Exosurf) Aerosolized Dose: 112 mg DPPC/kg/day Continuous Duration: 120 h	No difference in mortality, duration of MV and length of ICU stay	Poor alveolar deposition Only included sepsis induced ARDS patients No surfactant proteins
Gregory 1997 [62]	N = 59 Mixed ARDS	Bovine natural surfactant (Survanta) Intratracheal instillation Dose: 50 mg or 100 mg/kg Duration: 96 h	No difference in mortality Improved oxygenation with 100 mg 4 doses.	Dose finding study not powered for robust outcomes
Spragg 2003 [73]	N = 40 Mixed ARDS	rSP-C based synthetic surfactant (Venticute) Intratracheal instillation Dose: 0.5 mL/kg or 1 mL/kg (1 mL = 1 mg of rSP-C and 50 mg PL) Duration: 24 h	Feasible and safe to deliver No difference in oxygenation No difference in VFDs No change in surface tension lowering function at 48 h No detectable exogenous surfactant at 120 h	No improved surface activity at 48 h Short duration of treatment
Spragg 2004 [74]	N = 448 Mixed ARDS	rSP-C based synthetic surfactant (Venticute) Intratracheal instillation Dose: 1 mL/kg (1 mL = 1 mg of rSP-C and 50 mg PL) Duration: 24 h	No difference in mortality No change in VFDs Better oxygenation Post Hoc analysis: Direct ARDS patients had better outcomes	Short duration of therapy Improved oxygenation is not sustained
Kesecioglu 2009 [77]	N = 418 Mixed ARDS	Natural freeze-dried porcine surfactant (HL-10) Intratracheal instillation Dose: 600 mg/kg Duration: 36 h	No improvement in mortality or oxygenation Increased adverse events	Lack of preclinical data Large bolus instillation Terminated early
Spragg 2011 [76]	N = 843 Direct ARDS	rSP-C based synthetic surfactant (Venticute) Intratracheal instillation Dose: 1 mL/kg (1 mL = 1 mg of rSP-C and 50 mg PL) Duration: 96 h	No improvement in mortality or oxygenation	Partial inactivation during resuspension process. Terminated early
Willson 2015 [79]	N = 308 Direct ARDS	Natural calf Calfactant (Pneumasurf) Intratracheal instillation Dose: 30 mg/cm height Duration: 12–24 h (Most < 12 h) N = 151 had 1 dose (0 h) N = 78 had 2 doses (12 h) N = 3 had 3 doses (24 h)	No improvement in mortality or oxygenation No difference in ICU LOS, hospital LOS or VFDs	Short duration of therapy

ARDS: acute respiratory distress syndrome; DPPC: dipalmitoyl phosphatidylcholine; ICU: intensive care unit; LOS: length of stay; MV: mechanical ventilation; PL: phospholipid; rSP-C: recombinant surfactant protein C; VFD: ventilator free days.

## Data Availability

Not applicable.

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
