# Peer review of "Pulmonary Surfactant in Adult ARDS: Current Perspectives and Future Directions"

_diagnostics, 2023, doi:10.3390/diagnostics13182964_

Round 1
Reviewer 1 Report
Thank you for your nice overview about surfactant replacement therapy. For adult ARDS it looks very completely and your expertise in this field is outstanding. But of course as a pediatric intensivist I have some additional comments.
1.) you mentioned your paper in Critical Care 2012 as reference 94 . You should refere more in detail to the newer research papers from 2012 until today. it seems that studies nowadys focus on inhaled therapy and further studies also should.
2.) May be we should discuss not only about subgroups by diagnosis or direct or indirect lung injury, but also by severity of ARDS in the presented studies and as a possible design for future studies.
In practice there was a shift to more High flow therapy and more non-invasive management of early stages of ARDS. The threshold for invasive ventilation is not clearly worked out. There exist some expert opinion that moderate ARDS with paO2/FiO2 below 150 at least does need intubation.
3.) You nicely described the situation of RDS in premature infants, but you did not mention that already in near term infants the effect of surfactant treatment is limited and only combined outcome avoidance of ECMO and death was significant.
El Shahed AI, Dargaville P, Ohlsson A, Soll RF. Surfactant for meconium aspiration syndrome in full term/near term infants. Cochrane Database Syst Rev. 2007 Jul 18;(3):CD002054.
Willson DF, Jiao JH, Bauman LA, Zaritsky A, Craft H, Dockery K, Conrad D, Dalton H. Calf's lung surfactant extract in acute hypoxemic respiratory failure in children. Crit Care Med. 1996 Aug;24(8):1316-22.
De Luca described the perspectives in surfactant therapy for neonatal and pediatric ARDS: Biomed J. 2021 Dec; 44(6): 654–662.
4.) And in Pediatric ARDS there are studies or a review, which describes the results of surfactant in children.
Willson DF, Thomas NJ, Tamburro R, Truemper E, Truwit J, Conaway M, Traul C, Egan EE; Pediatric Acute Lung and Sepsis Investigators Network. Pediatric calfactant in acute respiratory distress syndrome trial. Pediatr Crit Care Med. 2013 Sep; 14(7):657-65.
Thomas NJ, Guardia CG, Moya FR, Cheifetz IM, Markovitz B, Cruces P, Barton P, Segal R, Simmons P, Randolph AG; PALISI Network. A pilot, randomized, controlled clinical trial of lucinactant, a peptidecontaining synthetic surfactant, in infants with acute hypoxemic respiratory failure. Pediatr Crit Care Med. 2012 Nov;13(6):646-53.
Tamburro RF, Kneyber MC; Pediatric Acute Lung Injury Consensus Conference Group. Pulmonary specific ancillary treatment for pediatric acute respiratory distress syndrome: proceedings from the Pediatric Acute Lung Injury Consensus Conference. Pediatr Crit Care Med. 2015 Jun;16(5 Suppl 1):S61-72.
5.) The immune systeme may play a role with less inactivation and less phagcytosis of surfactant in immature infants and therefore better acute effects.
6.) And I have a last idea: May be if it is possible to use X Rays or maybe better CT Scans to plan the strategy of surfactant application. eg. if there is a lot of atelectasis that cannot be recruited by optimized ventilation we should check whether regional application of surfactant improves aeration of the lungs. Surfactant in regions with overdistension should be avoided.
You may mention that in further studies protocols about ventilator settings should be comparable.
Author Response
Thank you for the reviewers comments. Please see our response.
Thank you for your nice overview about surfactant replacement therapy. For adult ARDS it looks very completely and your expertise in this field is outstanding. But of course as a pediatric intensivist I have some additional comments.
1.) you mentioned your paper in Critical Care 2012 as reference 94 . You should refere more in detail to the newer research papers from 2012 until today. it seems that studies nowadys focus on inhaled therapy and further studies also should.
Response: We have now included addtional references.
2.) May be we should discuss not only about subgroups by diagnosis or direct or indirect lung injury, but also by severity of ARDS in the presented studies and as a possible design for future studies.
Response: We have now introduced two sentences in the section 5.6. We have modified the paragraph as follows to address this comment.
"So far, all ARDS clinical trials are conducted on patients with moderate ARDS. The post-hoc analysis of rSP-C clinical trials suggest that severe direct lung injury from pneumonia or aspiration of gastric contents may be more responsive to surfactant replacement than indirect causes such as sepsis [75]. The most severe patients defined by the PaO2/FiO2 ratio of up to 100mmHg had the most improvement in oxygenation from surfactant replacement [75]".
"Future studies should explore prolonged exogenous surfactant replacement focusing on severe direct lung injury traits."
In practice there was a shift to more High flow therapy and more non-invasive management of early stages of ARDS. The threshold for invasive ventilation is not clearly worked out. There exist some expert opinion that moderate ARDS with paO2/FiO2 below 150 at least does need intubation.
Response: We agree with the reviewer that there are no definite oxygenation parameters help with initiation of mechanical ventilation. However, mechanical ventilation in the context of ARDS is associated with significant mortality and morbidity in comparison to non-invasive measures. The need for mechanical ventilation is a complex decision and should not be based on single variable such as oxygenation.
3.) You nicely described the situation of RDS in premature infants, but you did not mention that already in near term infants the effect of surfactant treatment is limited and only combined outcome avoidance of ECMO and death was significant.
El Shahed AI, Dargaville P, Ohlsson A, Soll RF. Surfactant for meconium aspiration syndrome in full term/near term infants. Cochrane Database Syst Rev. 2007 Jul 18;(3):CD002054.
Willson DF, Jiao JH, Bauman LA, Zaritsky A, Craft H, Dockery K, Conrad D, Dalton H. Calf's lung surfactant extract in acute hypoxemic respiratory failure in children. Crit Care Med. 1996 Aug;24(8):1316-22.
De Luca described the perspectives in surfactant therapy for neonatal and pediatric ARDS: Biomed J. 2021 Dec; 44(6): 654–662.
Response: Thank you. This review was mainly aimed to address adult ARDS. We have now included a sentence to address this comment (line 584-588).
"Similarly, the effect of surfactant replacement in pediatric ARDS are limited [105,106]. This difference in the response to exogenous surfactant replacement between nRDS and other disease cohorts may be due to the complexity of pathological processes including the progressive development of immune system leading to inflammation mediated increased surfactant catabolism in adult and pediatric ARDS".
4.) And in Pediatric ARDS there are studies or a review, which describes the results of surfactant in children.
Willson DF, Thomas NJ, Tamburro R, Truemper E, Truwit J, Conaway M, Traul C, Egan EE; Pediatric Acute Lung and Sepsis Investigators Network. Pediatric calfactant in acute respiratory distress syndrome trial. Pediatr Crit Care Med. 2013 Sep; 14(7):657-65.
Thomas NJ, Guardia CG, Moya FR, Cheifetz IM, Markovitz B, Cruces P, Barton P, Segal R, Simmons P, Randolph AG; PALISI Network. A pilot, randomized, controlled clinical trial of lucinactant, a peptidecontaining synthetic surfactant, in infants with acute hypoxemic respiratory failure. Pediatr Crit Care Med. 2012 Nov;13(6):646-53.
Tamburro RF, Kneyber MC; Pediatric Acute Lung Injury Consensus Conference Group. Pulmonary specific ancillary treatment for pediatric acute respiratory distress syndrome: proceedings from the Pediatric Acute Lung Injury Consensus Conference. Pediatr Crit Care Med. 2015 Jun;16(5 Suppl 1):S61-72.
Response: Please see previous response.
5.) The immune systeme may play a role with less inactivation and less phagcytosis of surfactant in immature infants and therefore better acute effects.
Response: We agree with the reviewer and included a sentence to address this comment (line 585-588).
6.) And I have a last idea: May be if it is possible to use X Rays or maybe better CT Scans to plan the strategy of surfactant application. eg. if there is a lot of atelectasis that cannot be recruited by optimized ventilation we should check whether regional application of surfactant improves aeration of the lungs. Surfactant in regions with overdistension should be avoided.
Response: We have introduced a paragraph in the section 5.5 to address this comment (line 365-374).
You may mention that in further studies protocols about ventilator settings should be comparable.
Response: We have introduced a sentence in section 8.0 (line 538-540).
Reviewer 2 Report
The author has summarized the association between pulmonary surfactants and ARDS; this manuscript provides a good summary, emphasizing the potential value of pulmonary surfactants in treating ARDS. With the following improvements, the quality of the manuscript will be further enhanced:
1. The author should provide the title with more substance rather than a simple description.
2. It is recommended to have a separate section discussing PS components such as SP-A and SP-D.
3. Increase insights into the molecular mechanisms.
4. Improve Figure 1 to make it look less monotonous.
5. Strengthen the conclusion section of the manuscript; currently, the manuscript does not provide us with a good summary of the entire paper.
Minor editing of English language required.
Author Response
Thank you for the reviewer's comments and please see our response below.
The author has summarized the association between pulmonary surfactants and ARDS; this manuscript provides a good summary, emphasizing the potential value of pulmonary surfactants in treating ARDS. With the following improvements, the quality of the manuscript will be further enhanced:
1. The author should provide the title with more substance rather than a simple description.
Response: We have modified the title as Pulmonary surfactant in adult ARDS: current perspectives and future directions.
2. It is recommended to have a separate section discussing PS components such as SP-A and SP-D.
Response: The review as not aimed to provide in-depth details of surfactant composition and function, including SP-A and SP-D. To address this issue we have added a sentence in the section 1.1 (lines 55-56).
3. Increase insights into the molecular mechanisms.
Response: We have introduced a new section (3.0) and a paragraph (lines 197-219).
4. Improve Figure 1 to make it look less monotonous.
Response: We have modified the figure now.
5. Strengthen the conclusion section of the manuscript; currently, the manuscript does not provide us with a good summary of the entire paper.
Response: We have modified this section to reflect the manuscript.
Reviewer 3 Report
Congratulations on your brilliant work.
this study could provide scientific insight into the relevant field and translated. A comprehensive understanding in future work.
Author Response
Thank you for the reviewer's comments.
Congratulations on your brilliant work.
this study could provide scientific insight into the relevant field and translated. A comprehensive understanding in future work